# A tool for functional brain imaging with lifespan compliance

Ryan M. Hill[1], Elena Boto[1], Niall Holmes[1], Caroline Hartley [2], Zelekha A. Seedat[1], James Leggett[1], Gillian Roberts[1], Vishal Shah[3], Tim M. Tierney[4], Mark W. Woolrich[5], Charlotte J. Stagg [5], Gareth R. Barnes [4], Richard Bowtell[1], Rebeccah Slater [2] & Matthew J. Brookes[1]*

The human brain undergoes significant functional and structural changes in the first decades of life, as the foundations for human cognition are laid down. However, non-invasive imaging techniques to investigate brain function throughout neurodevelopment are limited due to growth in head-size with age and substantial head movement in young participants. Experimental designs to probe brain function are also limited by the unnatural environment typical brain imaging systems impose. However, developments in quantum technology allowed fabrication of a new generation of wearable magnetoencephalography (MEG) technology with the potential to revolutionise electrophysiological measures of brain activity. Here we demonstrate a lifespan-compliant MEG system, showing recordings of high fidelity data in toddlers, young children, teenagers and adults. We show how this system can support new types of experimental paradigm involving naturalistic learning. This work reveals a new approach to functional imaging, providing a robust platform for investigation of neurodevelopment in health and disease.

[1] Sir Peter Mansfield Imaging Centre, School of Physics and Astronomy, University of Nottingham, University Park, Nottingham NG7 2RD, United Kingdom. [2] Department of Paediatrics, University of Oxford, John Radcliffe Hospital, Oxford OX3 9DU, United Kingdom. [3] QuSpin Inc., 2011 Cherry Street, Unit 112, Louisville, CO 80027, USA. [4] Wellcome Centre for Human Neuroimaging, UCL Institute of Neurology, University College London, 12 Queen Square, London WC1N 3AR, United Kingdom. [5] Oxford Centre for Human Brain Activity (OHBA), Wellcome Centre for Integrative Neuro-Imaging, Department of psychiatry, University of Oxford, Warneford Hospital, Oxford OX3 7JX, United Kingdom. *email: matthew.brookes@nottingham.ac.uk

The fundamental building blocks of human cognition are laid down within the first decades of life. In the early years, capabilities, such as movement, language and social interaction develop; during later years (late childhood and adolescence), cognitive and attentional mechanisms further evolve alongside finer motor skills. However, relatively little is known about maturation of brain function during these critical times. Brain imaging provides a wealth of information on both structure and function, but limitations in technologies, such as magnetic resonance imaging (MRI) or magnetoencephalography (MEG), are exposed when attempting to investigate neurodevelopment, in particular during naturalistic tasks. This is for two reasons: firstly, scanner technology tends to be optimised for adults, and the difference in the head-sizes of infants and adults is problematic for some systems. Younger participants are also less able to tolerate the imaging environment and excessive movement negatively impacts data quality. In clinical practice, participants are often sedated to prevent excessive movement; however, this approach is not feasible when studying brain function. These limitations prohibit us from asking key neurodevelopmental questions in younger participants. Secondly, the unnatural environments presented by most neuroimaging modalities impose significant limitations on the type of experimental paradigm that participants can undergo. For example, restricted head movements make it hard to interact naturally with engaging tasks, such as computer games (as, for example, complete immersion can result in involuntary movement). Similarly, restricted movement, coupled with limited space inside the scanner, make it hard to allow participants to undertake natural learning experiences, such as playing a musical instrument. To date, the best functional imaging solutions, both for measurement in infants and for naturalistic experimentation, involve wearable technologies, such as electroencephalography (EEG). However, EEG is limited, as it is highly susceptible to movement artifact[1–4] and lacks spatial precision[5]. If we are to understand the developmental trajectory of human brain function, and its perturbation by disorders, then new technologies are required.

Here, we aimed to solve this problem by creating a wearable MEG system with lifespan compliance. MEG measures the magnetic fields generated outside the head by neural current flow[6,7], and in this way offers measures of brain electrophysiology with high spatiotemporal precision[5]. Traditional (superconducting) MEG sensors require cooling inside a large cryogenic dewar, meaning systems cannot adapt to head shape/size and require participants to keep still during data acquisition. However, recently developed optically pumped magnetometers[8–11] (OPMs) offer a means to measure the small magnetic fields generated by the brain[12–18]; they are small and lightweight, and can be positioned flexibly on the scalp surface. This means that an OPM-MEG system can be adapted to any head shape/size. Furthermore, if background magnetic fields are appropriately nulled[19], OPMs can be mounted on the head and participants can move during scanning. Our initial work[1] showed proof of principle of OPM-MEG, with high fidelity data captured whilst a subject moved freely. However, the prototype system used heavy and costly 3D-printed helmets for sensor mounting. Those helmets, which were customised for each individual participant, are not appropriate for use in infants and young children. Further, our inability to adapt the helmets to different head shapes/sizes meant the high cost precluded their use in large cohort studies. Thus our initial system was impractical for deployment.

In this communication, using simulations to theoretically assess signal strengths[20], we describe a redesign of our original system to incorporate a simple and ergonomic helmet, which is practical in children and offers genuine lifespan compliance. We solve the significant problem of OPM-sensor localisation in order to that 3D images of electrophysiological changes in the brain can be generated. We also demonstrate how our redesigned system can be used to scan individuals across the lifespan, including children as young as 2 years of age. Finally, in adolescents and young adults, we show how this system supports the introduction of naturalistic experimental paradigms, enabling investigation of the neural substrates underlying motor control.

## Results

**A paediatric MEG system (Maternal Touch).** We first aimed to test the feasibility of our approach by recording brain activity in response to maternal touch in two children aged 2 and 5 years. Twelve OPMs (QuSpin Inc.) were mounted bilaterally above the somatosensory cortex in a modified bicycle helmet design. Simulation studies (below) demonstrate that this simple design represented an excellent balance between signal strength and ergonomics, and was well tolerated by the children. Indeed, the children had the opportunity to wear the helmet at home before the scanning session and even to wear a replica on a bike ride, in order to familiarise themselves with the helmet and reduce anxiety during scanning. The helmet was altered to remove magnetic material and mounts were added to rigidly house the OPMs. Each OPM is a self-contained unit in which a 795-nm, circularly polarised laser beam is shone through a cell containing rubidium vapour[9]. In the absence of magnetic fields, the atoms become spin-polarised via optical pumping, and the cell becomes transparent to the laser. However, when a magnetic field is applied perpendicular to the beam the atoms undergo Larmor precession and a measurable drop in light transmission occurs, providing a sensitive marker of local neuromagnetic field (for a review see ref. [21]). The helmet (weighing ~400 g) was integrated with four reference OPMs (to monitor environmental fields), and a coil system which nulled the background field[1,19], allowing the child to move their head whilst being scanned.

OPM-MEG data were recorded during a sensory task, in which the child's mother gently stroked the thenar eminence on the palm of the hand with a soft brush. Stimulation for 2 s was followed by 3 s rest in 50 trials, across which data were averaged. The children watched a television programme throughout. An adult (aged 24) was also scanned using the same task (and a larger helmet). Movement of the head was tracked using an infrared camera; the maximum head translations/rotations were 2 cm/6°, 2.5 cm/13° and 2 cm/6° for the 2-, 5- and 24-year-old, respectively. The experimental setup is shown in Fig. 1a. Clear changes in brain activity were recorded in response to stimulation using synthesised gradiometers (Fig. 1b). A reduction in the mu/beta amplitude was observed during stimulation in all three participants, as would be expected[22,23].

In order to generate the images showing the cortical origin of the beta band effect, accurate information on the locations and orientations of the sensors relative to the brain is required. This problem was solved using a combination of X-ray computed tomography of the helmet, an infrared motion tracking camera, and a 3D digitisation of the participant's head and face (see below). This combination enabled us to accurately identify the sensor locations/orientations relative to the participant's head, allowing coregistration of sensor locations to brain anatomy. A beamformer technique[24] was then used to localise sources of beta modulation. Using this approach, the largest change in beta activity was identified in the contralateral primary somatosensory cortex in all participants (see Fig. 1c).

**An interactive motor paradigm (Robin Hood).** Next, we aimed to show that the same system could be used in older children, alongside a naturalistic (and engaging) motor paradigm. A single

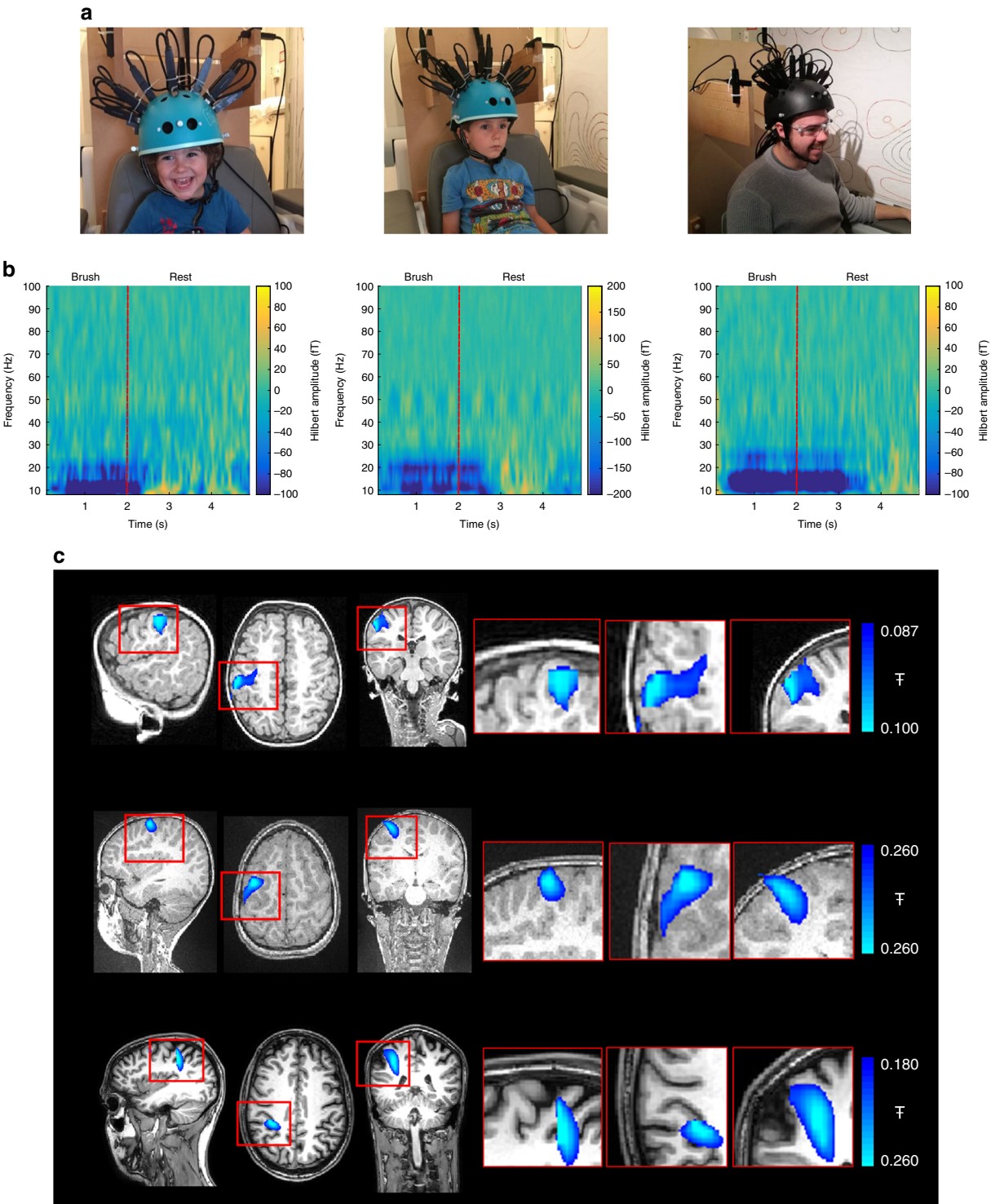

**Fig. 1** A paediatric MEG system: **a** Experimental setup for three participants age 2- (left), 5- (centre) and 24-years (right). OPMs, housed in a modified bike helmet, measured the MEG signal. **b** Time-frequency spectra from a single (synthesised gradiometer) channel. Changes in neural oscillations are shown; blue indicates a reduction in oscillatory amplitude relative to baseline; yellow indicates an increase. Note reduction in beta (13–30 Hz) and mu (8–13 Hz) amplitude. **c** The spatial signature of beta modulation during the period of tactile stimulation (0 s < t < 2 s) (blue overlay)

14-year-old female participant took part in the experiment. Our Robin Hood paradigm (See Fig. 2a) comprised a set of targets displayed on a screen. A cross hair smoothly moved across the screen at a set speed, and at specific times passed over a target. The subject had to move their right index finger in order to shoot an arrow at the target. When a shot was fired the position of the arrow was recorded as a black marker on the screen, and the participant was able to score points by hitting the targets. We reasoned that this paradigm, which took the form of a motion-controlled computer game, would be engaging for teenagers. Control of the game via movement was enabled by an infrared reflective marker placed on the right index finger. Movement was

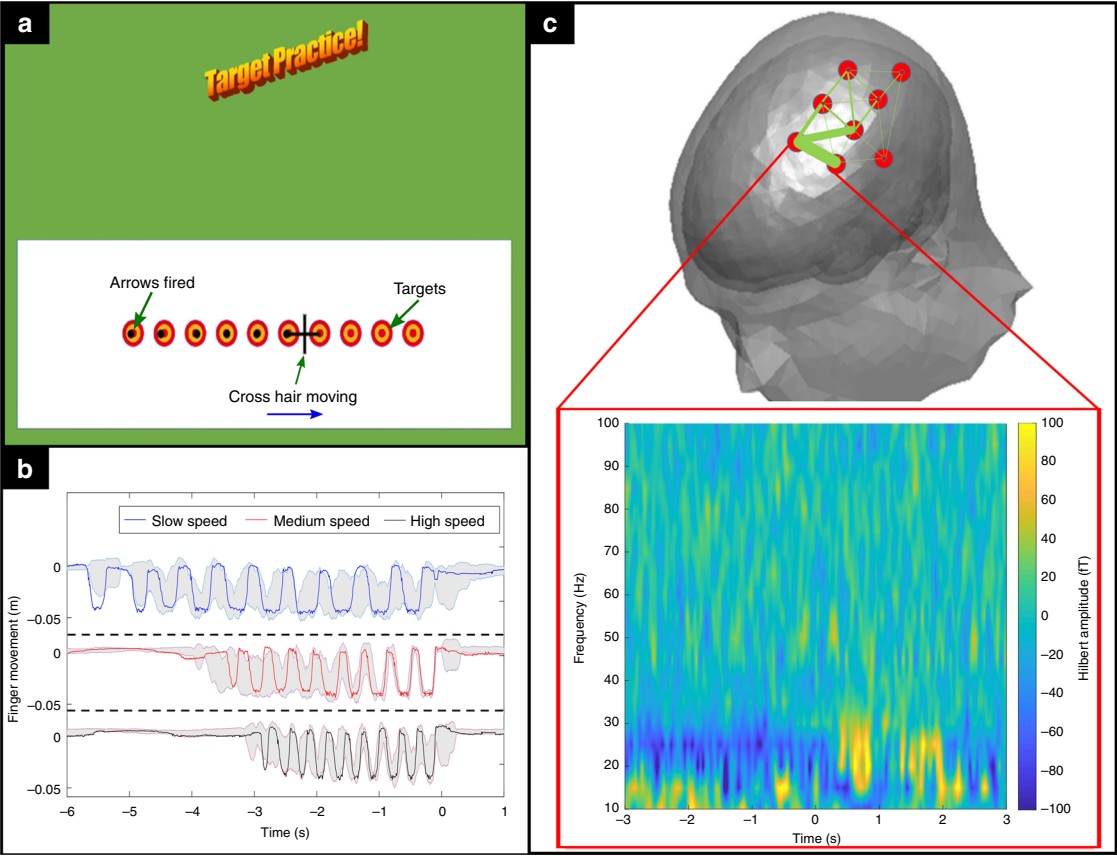

**Fig. 2** An interactive motor paradigm: **a** The paradigm: a cross hair moves across the screen passing targets. The participant moves their finger in order to shoot an arrow at the target. **b** Measured finger movements for three different speeds of cross hair movement. The shaded area shows the average and standard deviation across trials whilst the solid line shows a single example trial. **c** Beta responses in a single 14-year-old participant. The upper plot shows a schematic diagram of the positions of the OPMs; the thickness of the line denotes the strength of the change in beta amplitude for all gradiometers. The lower plot shows a time-frequency spectrogram from the most anterior gradiometer pair, with time zero representing the offset of movement, measured from the recorded finger movement; all trials have been averaged

tracked by a 3D camera and fed back, in real time to the paradigm, as well as being recorded by the OPM-MEG system. The game slowly got harder during the experiment with the cross hairs moving more quickly.

The recorded finger movement is shown in Fig. 2b; note that as the task progresses the movement required becomes faster. The finger movement can be characterised accurately, and used to trigger data analysis. In this case we synchronised data analysis (time $t = 0$) to the measured offset of motion. The sensor-level MEG results are shown in Fig. 2c. In the upper plot, the line thickness indicates which gradiometer pairings showed the largest beta band response. The beta response itself is shown below in the time-frequency spectrogram, with the expected desynchronization during movement shown clearly[23,25,26].

**Naturalistic motor learning (Ukulele).** As a final demonstration of the versatility of our system we aimed to scan an adult executing a naturalistic task. A single 24-year-old female took part in an experiment where she was asked to play a musical instrument. Specifically, they were asked to learn to play a sequence of five chords on a ukulele. The chords were visually projected onto a screen, and she was given 5 s to complete the sequence. This was repeated 40 times. The experimental setup is shown in Fig. 3a with the inset image showing the visual presentation. Figure 3b shows measured head and right hand movement, as well as the recorded sound made by the instrument. A single example trial is shown (in which the subject only

managed to play three of the five chords). In this naturalistic motor learning task, the participant had to make natural head movements, as they looked first at the chord pattern on the screen, and then at the frets on the ukulele to form the chord. These head movements were measured to be $4.73 \pm 1.21$ cm (translation) and $28.1° \pm 6.71°$ (rotation) (average ± standard deviation in the dominant direction across trials). Despite the substantial head movement that was required to complete the task, clear electrophysiological responses are observed in the time-frequency spectrum shown in Fig. 3c, including both the expected beta[23] and gamma[27] band responses.

**System design**. Measuring electrophysiological brain activity in each of the three experimental paradigms was possible due to the careful design of generic helmets which hold the OPM sensors. Previous demonstrations (all in adults) used 3D-printed helmets[1,28] that were individualised for each participant. Whilst this approach maximises signal quality by ensuring optimal sensor placement on the scalp surface, it is expensive and inappropriate for use with children, since the helmets are heavy, uncomfortable and intimidating. Fabrication of a generic helmet, balancing practicality against sensitivity, is key to realising the potential of OPM-MEG for use across the lifespan and in large cohorts. We identified an appropriate balance of these two factors by simulating MEG signal strength in a 2-year-old child, using a hypothetical array of 81 OPMs housed in four different helmet designs: (A) a 3D-printed bespoke helmet; (B), a generic helmet

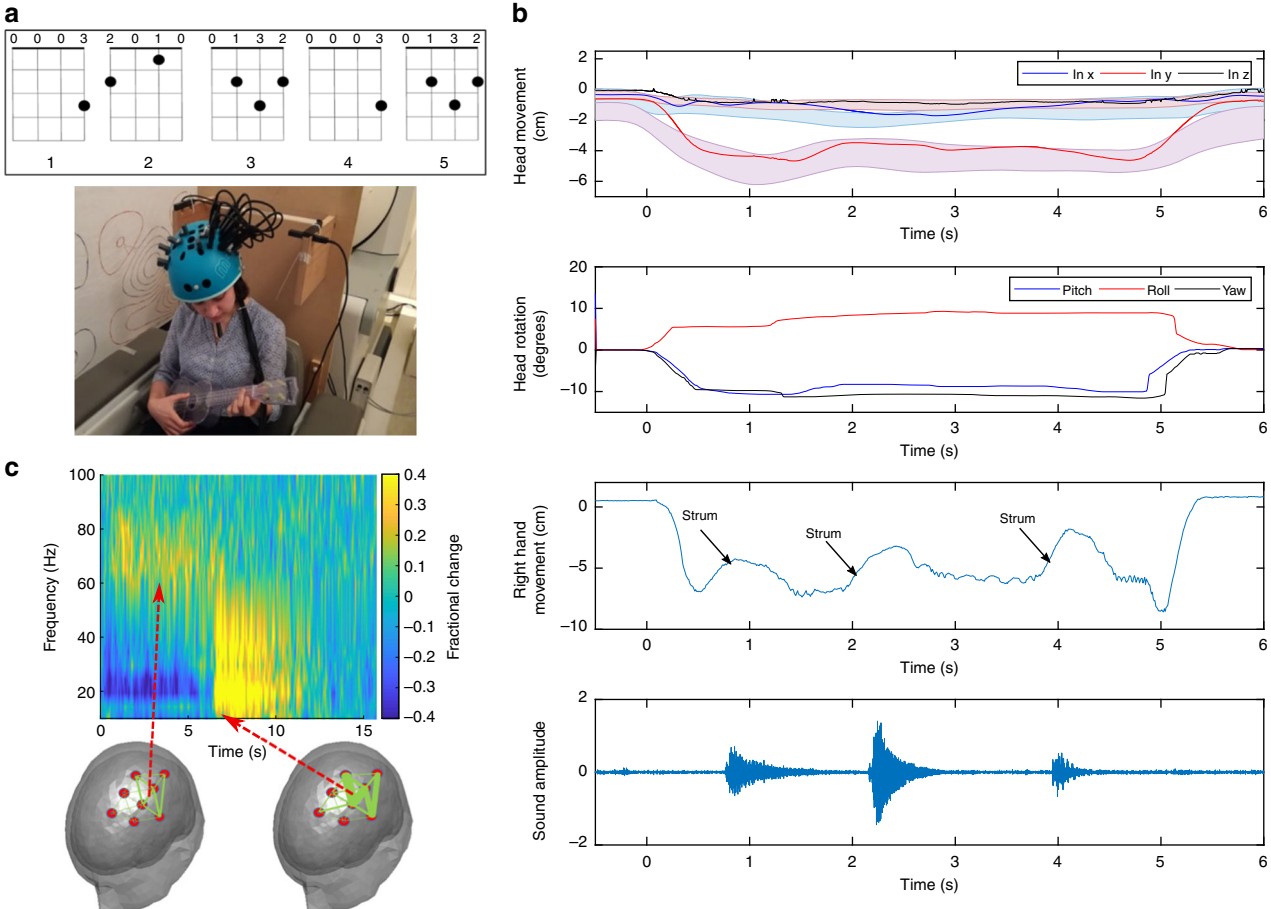

**Fig. 3** Naturalistic motor learning: **a** Experimental setup with a participant playing a musical instrument. Inset image shows what the participant saw on the screen. **b** Measured head and right hand movement showing the timing of when the ukulele was strummed. The lower trace shows the sound generated. **c** Schematic diagram of sensor placement; the line thickness denotes the strength of the effect in the gamma band (left) and the beta band (right). Time-frequency spectrum shows the neural oscillatory response from the gradiometer pair indicated by the dotted red line

designed to fit 95% of 2-year-olds; (C) a standard child bike helmet and (D) a cryogenic MEG system (see Fig. 4). Although the bike helmet results in a drop in signal strength compared to the bespoke design, it is simple, cheap, practical, suited to children and it vastly outperforms cryogenic systems, making it an ideal compromise.

**Coregistration procedure**. Having mounted the OPMs rigidly in the helmet, a further challenge is knowing where their sensing volumes are positioned in relation to the brain anatomy. This is critical if images showing source localised brain activity are to be derived. Here this problem was solved by a three-step process. First, a volumetric X-ray computed tomography scan of the helmet, with sensors in situ, was acquired yielding a 3D digital representation at a resolution of 162.9 μm. (Note—only imitation OPMs were used in the actual X-ray to avoid damage to the real sensors) This gave the locations of the OPMs relative to each other, and in relation to known markers on the helmet. Second, our motion tracking camera was used to monitor the position of the helmet relative to a pair of glasses worn by the subject. This provides ~1-mm per degree precision; it allows knowledge of the location of the helmet on the head and further, enabled us to test whether the helmet moved, relative to the head, during the scan. Finally, the subject's head shape (scalp and face) was digitised using a 3D digitiser in a co-ordinate system defined relative to the markers on the glasses. Combining data from the CT scan, camera and digitisation facilitated a complete coregistration of the

location and orientation of the OPMs relative to the head. This process is shown schematically in Fig. 5a.

To further validate this coregistration procedure, three adults underwent a finger movement paradigm[1]. A single trial comprised 2 s of right index finger abduction followed by 3 s rest. The experiment consisted of 100 trials. A 24 OPM array, which covered the left and right primary motor cortices, was used. Results are shown in Fig. 5b. Note that, despite relatively sparse sensor coverage, beta modulation was well localised to contralateral motor cortex. Quantitatively, the peaks in the beta response were found at MNI coordinates (−44−16 56) mm, (−36−22 48) mm and, (−42−6 52) mm for the three subjects. According to the Oxford-Harvard probabilistic atlas, all three coordinates fall in pre-central-sulcus, as would be expected for a motor response. In addition, all three subjects reconstructed data show the expected beta-band modulation with a clear movement related beta decrease during movement and post movement rebound on movement cessation.

## Discussion

This communication has described the practical implementation of a generic, lifespan-compliant, motion robust, OPM-MEG system. Using simulations we demonstrated that a simple, modified bicycle helmet offered a good compromise between high signal strength (getting sensors as close as possible to the head), and a practical and ergonomic design that could be deployed across all ages, and would be well tolerated by children. By

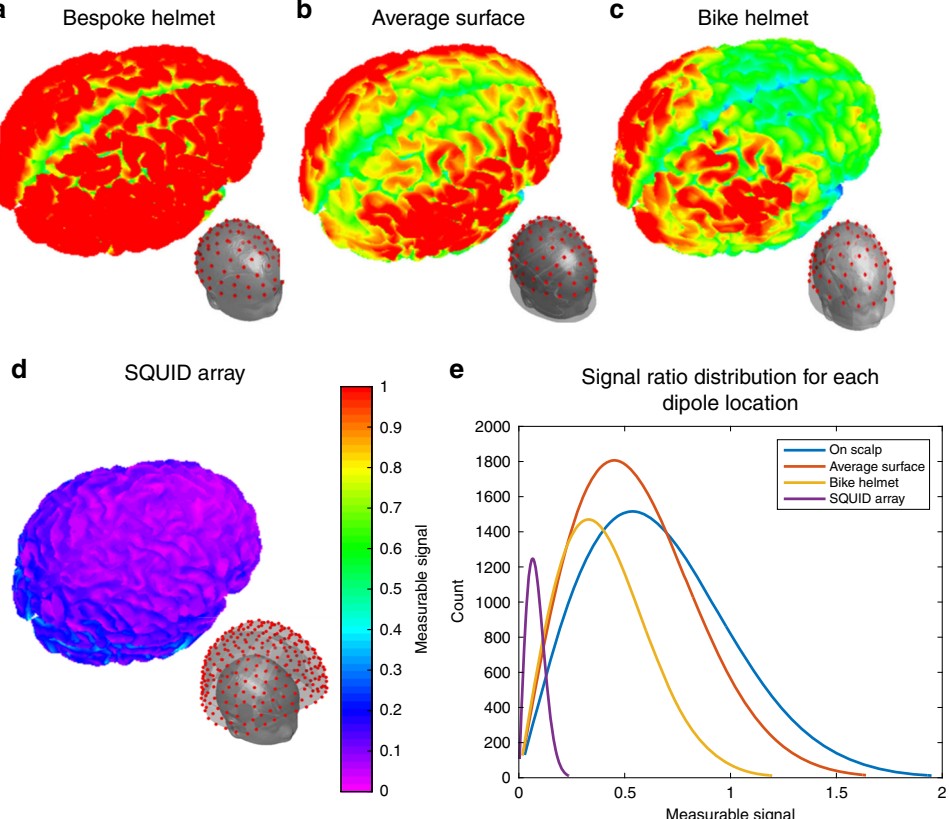

**Fig. 4** Simulated MEG helmet designs for a 2-year-old. Main figures show normalised representation of measurable signal across the brain. (Specifically a 1 nAm source dipole has been simulated at all cortical locations and we measured the Frobenius norm of the resulting sensor-level field pattern. These values were normalised by the average Frobenius norm when sensors are placed on the scalp surface—i.e. the values, which mostly scale between 0 and 1, represent signal quality relative to the bespoke helmet). Inset figures show the location of simulated MEG sensors. **a** Bespoke helmet. **b** Helmet to fit 95% of 2-year-olds. **c** Modified bike helmet. **d** Cryogenic MEG system. **e** Distribution of measurable signal values, across all dipole locations for each helmet type

introduction of a technique for coregistration between sensor locations and brain anatomy, we were able to generate 3-dimensional images of the spatial signature of changes in neural oscillations. We have used this system to provide a snapshot of how a single OPM-MEG device (with differently sized helmets) could be deployed to better understand neurodevelopment, in young children (aged 2- and 5-years), during adolescence (a 14-year-old) and in adults. Importantly these data show the feasibility of the technique to generate high quality data, even in a 2-year-old child (typically the hardest age to scan without sedation). Further we have shown that OPM-MEG enables us to measure brain activity during naturalistic motor paradigms.

Our movement controlled computer game represents a paradigm with significant potential for future use in understanding development of motor skills. Our game was simple—move a finger to fire an arrow—nevertheless it is easy to conceive more complex scenarios to test motor coordination (games along the lines of those developed for Microsoft X-box or Nintendo Wii might be possible). Further, motion tracking generates important and accurate data on how and when participant movement occurs. These data are particularly important in neurodevelopmental paradigms where infants and young children are less likely to follow instructions. Here, we only had a single tracking camera and this limited the number of markers that we were able to track simultaneously. However, there is no reason why an array of tracking cameras could not be used to measure a host of additional data; examples could range from measurements of posture and head position in babies (known to be important to

characterise attention and engagement) to whole-hand morphology to measure dexterity in older children. This kind of tracking would enable capture of the critical timing and motion parameters necessary to carry out neuroimaging analyses in naturalistic experiments—e.g. measuring when a child plays with a toy, and which fingers they are moving.

The idea of naturalistic stimulation was taken a step further (here in an adult) via a paradigm where a person who had no previous experience of playing a stringed instrument learnt to play a ukulele. Successfully recording the electrophysiological data required that the technology could cope with a range of motions including (i) rotating the head back and forth to look at the visual representation of the chords and the frets; (ii) moving the left hand to form the chords and (iii) moving the right hand to strum the strings. Despite head (and hence sensor) translations and rotations >7 cm and >30°, we were able to capture expected electrophysiological responses in both the beta and gamma frequency bands. This demonstration highlights the possibility to implement a naturalistic motor experiment in which the neural substrates underlying motor learning are probed.

There are some limitations to the current study. Most importantly, we used a limited number of OPMs, which meant only a fraction of the head was covered. Clearly a complete system would require whole head coverage (similar to that simulated in Fig. 4 with e.g. 81 sensors). There is no fundamental reason why the current system cannot be expanded by adding more OPMs. The only theoretical concern is cross-talk between sensors; however, this is a minor consideration since at the current sensor

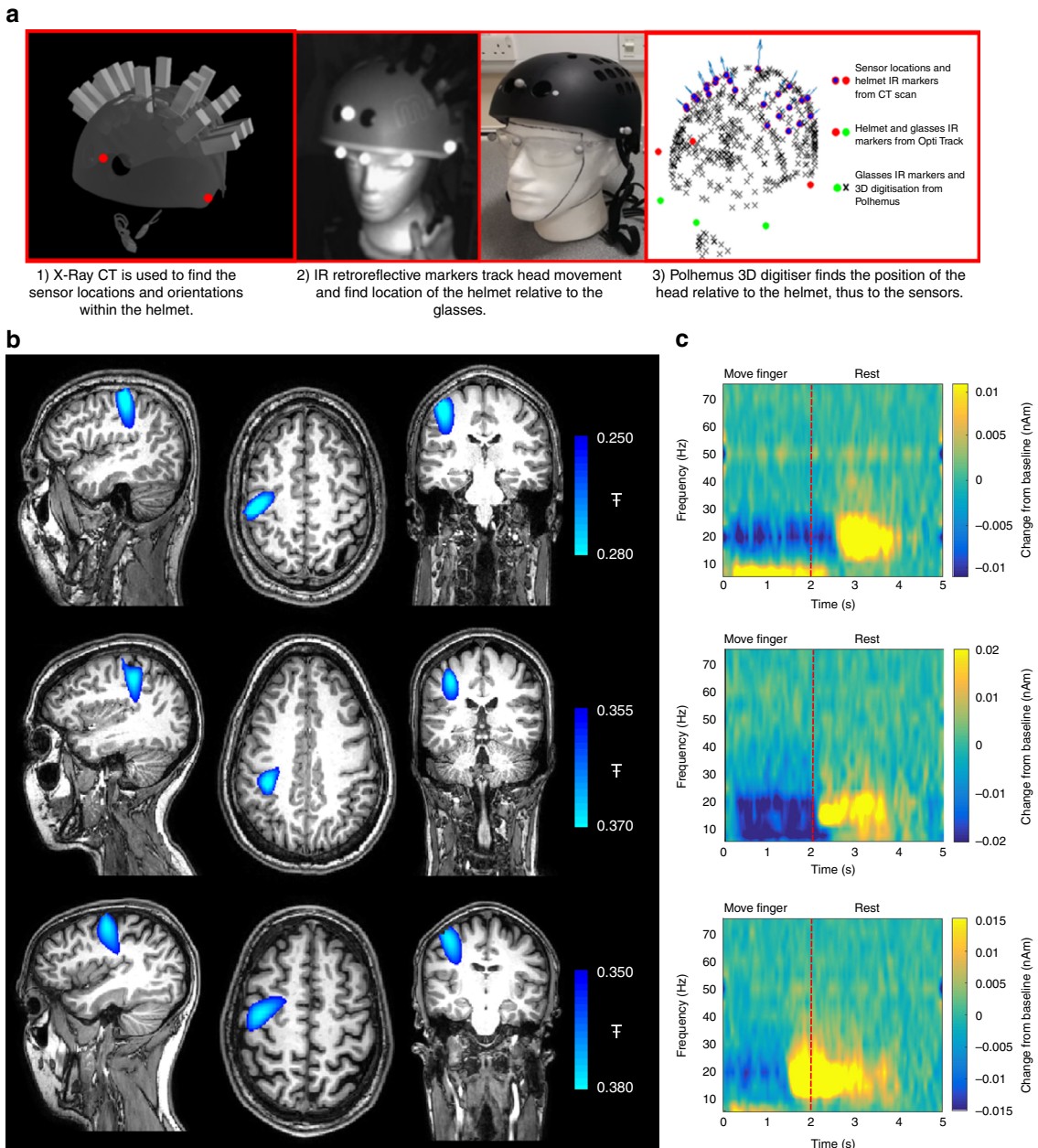

**Fig. 5** Coregistration procedure. **a** Schematic showing how sensor locations were determined relative to brain anatomy **b** Images showing the spatial signature of beta modulation; the largest peak was found in pre-central-sulcus (motor cortex) in all three subjects. **c** Time-frequency spectra, reconstructed using a beamformer at the location of maximum beta change (i.e. the peaks in Fig. 5b)

spacing cross-talk generates only ~3% error in sensor gain[1], which can be easily accounted for in source modelling. A more practical issue is system weight. Each OPM weighs 20 g and so an 81-channel system, including the helmet, would weigh ~1800 g. Further, the cables to each OPM currently weigh 33 g/m, and cables hanging from the head cause torque, which could become uncomfortable for long experimental durations, particularly for younger participants. However, these problems are solved by the availability of a new generation of commercial OPMs[2], which are smaller ($1.24 \times 1.66 \times 2.44$ cm$^3$), lighter (4 g) and use lighter cabling (3.3 g/m). This means that the weight of an 81-channel system could be as low as 500 g, with cable weight being negligible.

This study is also limited by the low number of participants included in each study paradigm. However, the neurophysiological effects that we demonstrate have previously been robustly observed in larger subject groups, in both adults[23] and children[22]. Our aim was to show proof of principle that these well documented responses could be seen clearly using OPM-MEG in individual participants. We have concentrated our findings on beta band effects at the expense of other important neuromagnetic signals (though we also see mu and gamma band effects). However, effects in other bands (e.g. theta) and evoked responses[1] have been shown in previous work and so there is no reason to suspect that similar responses would not be seen using the present system design. Finally limited scalp coverage and low sensor count means that source localisation is intrinsically constrained to specific brain areas (i.e. we will only localise sources under the sensors). Nevertheless we have shown the expected left hemisphere lateralisation of sensory and motor responses, and in our validation (Fig. 5) we see precise localisation to the hand area, in pre-central sulcus; this was with coverage using 24 sensors across the entire motor strip.

In summary we believe that OPM-MEG, with generic helmet design, paves the way for a new approach to neurodevelopmental research. We have designed a system with complete lifespan compliance that gives the opportunity for participants of all ages to take part in a range of tasks without the need to restrict participant movements. The human brain develops rapidly in the first decades of life, with unparalleled cortical plasticity. Our ability to measure how the brain responds and adapts to naturalistic events in the external environment, in children, adolescents and adults using a single system will provide mechanistic insights into how central nervous system function develops in early life. This approach will shape the future direction of developmental neuroimaging research.

## Methods

**Simulations for helmet design.** Successful fabrication of an OPM-MEG system requires the design of an optimised helmet, balancing practicality with sensitivity. To understand how helmet design impacts the measurable MEG signal, we undertook a set of simulations (similar to Boto et al.[20]).

A hypothetical 81-channel OPM array was used with sensor locations following, approximately, a 10–10 EEG arrangement. We also simulated a cryogenic array. The designs were as follows:

(1) A bespoke helmet, where OPMs are placed directly on the scalp surface, similar to 3D-printed helmets used in previous work[1,13]. The scalp surface was extracted from an anatomical MRI of a 2-year-old, selected at random from the Pediatric Brain Atlas[29]. Sensor locations were placed directly on this individual's head surface.

(2) A generic helmet based on the average head shape of a 2-year-old child. The average head shape was found by accessing 33 MRI anatomical images from the Pediatric Brain Atlas[29]. The scalp surface was extracted from each anatomical, and a k-dimensional tree[30] searched for the nearest neighbours between two of the scalp meshes. The midpoint between each pair of vertices was found, creating a new average surface. This process was repeated for each of the 33 anatomical images in turn, and the average surface updated. The vertices of the average surface were extended along their normals by the standard deviation of the individual surfaces, to generate a surface that would accommodate 95% of 2-year-old head shapes, and this became the inner surface of the simulated helmet. Having derived this inner surface, it was fitted, visually, to the MRI of the same 2-year-old used in (1).

(3) Sensors placed on the inner surface of a bicycle helmet, designed to fit children aged 1–6 years. The inner surface of a bike helmet was digitised using a Polhemus system and fitted, visually, to the scalp of the same 2-year-old used in (1) and (2). Simulated sensors were placed on the inner helmet surface.

(4) Finally, for comparison, the same 2-year-old's head was simulated inside a 275-channel adult CTF SQUID system; we simulated the head at the centre of the array, as well as resting against each of the sides and the top.

Each of these geometries was the basis for a simulation in which we compared the different MEG helmet designs to obtain the magnitude of the measurable signal as a function of brain location. In terms of signal magnitude, clearly the bespoke helmet represents the ideal scenario in which the sensors are as close to the brain as possible. However practical implementation of such a system would be difficult because 3D-printed helmets are intimidating, heavy and expensive, with a new helmet having to be made for each individual. In terms of practicality, the bicycle helmet is desirable; it is easily sourced, cheap and many participants can wear the same helmet. Moreover, it is something that children will be familiar with. In addition, children could take one home and practice with it at little cost.

The cortical surface was extracted from the anatomical MRI of a single 2-year-old, using Freesurfer (http://surfer.nmr.mgh.harvard.edu/). We simulated the fields from an array of dipoles (26,988 in total) that were distributed over the cortical surface with locally normal orientations. For each dipole we estimated the magnetic fields that would be measured at each sensor, for each of the four simulated geometries described above. The magnetic fields were derived using a spherical head model and the dipole solution first described by Sarvas[31] we assumed that the stand-off distance from the end of the sensor to the sensitive cell was 6 mm. For each dipole location, $\mathbf{j}$, the dipole moment was assumed to be 1 nAm, and the simulated forward field vector, $\mathbf{l_j}$, calculated. The Frobenius norm, $f_{\mathbf{j}}$, of the forward field was then calculated as:

$$f_{\mathbf{j}} = |\mathbf{l_j}|_F = \sqrt{\Sigma_{i=1}^{N_{ch}} |\mathbf{l_{ij}}|^2} \qquad (1)$$

where $N_{ch}$ is the number of channels (81 for the OPM arrays and 275 for the SQUID array) and $i$ indexes channel number. This resulted in 26,988 values of $f_{\mathbf{j}}$ for each of the four geometries, representing sensitivity of that specific geometry to all regions of the cortical surface. Following this, for each location, we divided the

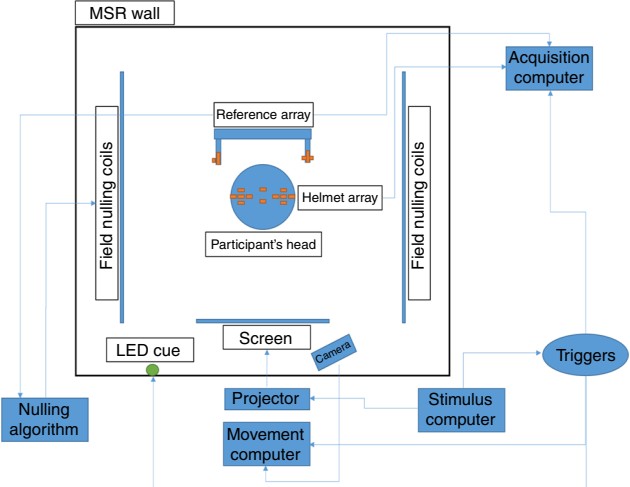

**Fig. 6** Schematic overview of the OPM-MEG system

Frobenius norm of the forward field by the mean Frobenius norm, over all superficial dipole locations (i.e. those generating the largest signal), for the on scalp array helmet; this resulted in, approximately, a 0 to 1 scale for the measurable signal amplitude with values of 1 being closest to the on scalp helmet. This allowed us to quantitatively measure the sensitivity lost when using more practical helmet designs. These simulated array sensitivities are shown in Fig. 4 of the main manuscript. Supplementary Fig. 1 shows differences found when moving the child's head within the CTF helmet.

**OPM-MEG system design and fabrication.** An overview of our system is shown in Fig. 6. The system used an array of QuSpin OPMs mounted in a bike helmet (head array) to measure the neuromagnetic field. A further four sensors were placed in an array located behind the head and used to record background magnetic interference (reference array). Each sensor produces an analogue voltage output proportional to magnetic field (scaling = 2.7 V nT$^{-1}$). Data were recorded from the sensors at a sampling frequency of 1200 Hz, using a 16-bit digital acquisition (DAQ) system comprising three NI-9205 modules, controlled using custom-written software in LabVIEW (National Instruments). The sensors were housed in a three layer magnetically shielded room (MSR) to reduce environmental interference; all control equipment was kept outside the room.

In order to enable free head movement during scanning, it is necessary to null any residual static (i.e. Earth's) magnetic field inside the MSR[1,19]. In the absence of such nulling, any movement of the sensors with respect to the background field will cause large artefacts (enough to send the sensors outside their dynamic range). To achieve this nulling, prior to data acquisition (but with the participant in place) we used the reference array to measure the ambient static field, and its three most prominent gradients. This information was fed into a control algorithm which outputs currents through an array of six electromagnetic coils[19] placed either side of the participant. The resulting magnetic fields from these coils cancelled the static magnetic field inside the MSR. Note that for a complete description of these bi-planar coils the reader is referred elsewhere[1,19]. During data acquisition, both the head array and the reference sensors were operated simultaneously with reference array data being used to characterise the background interference. Throughout the scan, stimulus delivery was controlled by a separate computer (running MATLAB), which also provided temporal markers to the acquisition computer.

The OPMs are compact, commercially available devices manufactured by QuSpin Inc. (Louisville, CO). The sensors have a sensitivity of <15 fT per √Hz, a dynamic range of ±1.5 nT, and a bandwidth of 0–130 Hz[9]. Each OPM consists of: a $3 \times 3 \times 3$ mm$^3$ glass cell containing $^{87}$Rb vapour heated to 150 °C; a semiconductor laser tuned to the D1-transition of $^{87}$Rb to spin polarise the vapour, optics for laser beam conditioning and a silicon photodiode for beam detection[8]. Three on-board coils generate three orthogonal fields, which can be used to null static field components within the cell. At zero magnetic field, the photo-detector signal is at a maximum. However, in the presence of small applied fields perpendicular to the laser beam, the atoms undergo Larmor precession, which decreases the transparency of the cell to the laser light. By monitoring the change in transmitted light intensity, changes in the magnetic field can be detected.

Two bicycle helmets (one child and one adult size) were modified to securely house the sensors. For the Finger Abduction paradigm 24 sensors were available; 12 sensors were available for the Maternal Touch experiment, and 8 for the Robin Hood and Ukulele experiments. Because of the limited number of sensors, we needed to ensure the sensors were positioned for optimal coverage of the region of interest of the brain (the hand area of primary somatosensory and motor cortices). A dipole was simulated at the centroid of the left sensory cortex, identified using the Automated Anatomical Labelling (AAL) atlas. Further simulated dipoles were

also placed within the same AAL parcel, with their magnitudes down-weighted in accordance with distance from the centroid. A forward field simulation (as above) projected a field pattern from these dipoles to the inner surface of the helmet, finding the optimal locations of slots within the helmet. Sensors were distributed around the area of highest signal. The tip of each sensor was flush with the inner surface of the helmet. In order to stabilise the sensors, small supporting blocks were screwed to the surface of the helmet, and the OPMs secured to these blocks. This ensured that the sensors could not move relative to the helmet (or each other) during data acquisition.

During normal operation each OPM cell is heated to 150 °C. Despite the sensor's thermal insulation, we wanted to ensure the temperature at the surface of the head was well controlled. To this end, we first wrapped a small piece of pyrolytic graphite (Panasonic Corporation Japan) around the end of each sensor and ran it out of the top of the bicycle helmet—this ensured that heat was efficiently conducted away. A piece of 0.5 mm thick Nomex thermal insulating sheet (DuPont, US) was also placed over the end of each sensor on the inner surface of the helmet. We conducted experiments to measure the temperature of the inner surface of the helmet during normal operation; the average temperature was 41.2 ± 0.8 °C (average over five measurements after 10 min). This was acceptable within the relevant guidelines[32].

A motion tracking camera was used during the experiments for several purposes: (1) to monitor the position of the helmet relative to the participant's head; (2) for coregistration of the MEG data to an anatomical MRI and (3) to trigger and record events during tasks (see below). The OptiTrack V120:Duo camera (NaturalPoint Inc., Corvallis) provides sub-1-millimetre and sub-1-degree precision optical tracking of a rigid body with a 120 Hz frame rate, using two cameras each with an array of 15 infrared LEDs. The OptiTrack uses the Motive software platform to record and track infrared (IR) retroreflective markers.

Additionally, the NatNet SDK software allowed control of the Motive software from a MATLAB script, enabling OptiTrack data to be synced to the MEG data. The NatNet software also allows marker information to be simultaneously streamed to MATLAB to control paradigms.

**Experimental method data acquisition**. All participants provided written informed consent for all experiments. The authors affirm that human research participants provided informed consent for publication of the images in Figs. 1a and 3a. For participants under 18 years old, written informed consent, both to their participation, and to the publication of the images, was given by their parents. All studies were approved by the University of Nottingham Medical School Ethics Committee. All MEG data were recorded using the OPM-MEG system described above. Participants were seated but allowed to move freely. Any visual cues to the participant were projected through a waveguide in the wall of the MSR onto a back-projection screen which was positioned ~40 cm in front of the participant.

For the maternal touch paradigm, three participants (male, ages 2, 5 and 24 years) took part in the study. 12 OPMs were placed over the left and right sensorimotor cortices. The stimulus involved an experimenter stroking the participant's right palm (thenar) using a soft brush for 2 s, followed by 3 s rest, repeated for 50 trials. Stimulation was cued by an LED. For the children, stimulation was applied by their mother. The children were also allowed to watch a television show during the scan.

In the Robin Hood paradigm, one participant (female aged 14, right handed) took part in the study. The participant played a game in which a crosshair moved over a series of ten targets, and the participant shot arrows by moving their right index finger. The speed of the crosshair changed throughout the experiment to alter the difficulty, with a slow trial taking 5 s for the cross hair to move across the screen, and a fast trial taking 3 s. Each trial was followed by 10 s of rest. There was a total of 30 trials split into three blocks of ten, with a 40 s rest period between each block. Eight OPMs were placed over the left sensorimotor cortex to measure the response from the right finger movement. The OptiTrack camera tracked a marker on the participant's finger throughout the task to provide real time feedback to the game. Finger movement data were recorded, and the firing of an arrow was triggered when finger movement velocity exceeded a threshold value.

As a side note, the participant had dental braces. This can be a contraindication to MEG because even small amounts of head movement relative to a fixed (cryogenic) sensor array would cause large artefacts due to the metal of the braces moving. However for OPM-MEG, since the braces are fixed in position relative to the sensors, this was not considered a problem. However, to accommodate this, the field nulling process was adapted: The background field was separated into two components: the remnant Earth's field in the room, $\mathbf{B_E}$, and the field due to the braces, $\mathbf{B_B}$. $\mathbf{B_E}$ is spatially locked to the MSR, whereas $\mathbf{B_B}$ is locked to the head, therefore in the reference frame of the OPMs only $\mathbf{B_E}$ changes with time if the head moves. Field nulling was therefore applied without the participant in the room, to null $\mathbf{B_E}$. Once the participant is brought in, the on-board-sensor coils were used to null $\mathbf{B_B}$.

For the Ukulele paradigm, one participant (female aged 24, right handed) took part in the study. A single trial involved an attempt to play five chords on a ukulele for 5 s followed by 10 s rest; the participant had to stop playing after 5 s regardless of whether all five chords were completed. Forty trials were performed and instructions on how to play the chords were presented visually. Again, 8 OPMs were placed over the left sensorimotor cortex to measure the response from the right (strumming) hand. The OptiTrack camera measured the head movement, as

well as movement of the strumming hand. An audio signal (from a microphone in the MSR) was also recorded. The Ukulele was modified slightly to remove metal components (including the tuning pegs, which were replaced with plastic).

In the finger abduction paradigm, three participants (two males, aged 24 and 28 years, and one female aged 27 years, all right handed) took part in the study. A grating appeared for 2 s, followed by a 3 s rest with a fixation cross shown throughout. During presentation of the grating, the participant was instructed to perform repeated abductions of their right (dominant) index finger, and to stop once the grating disappeared. Hundred trials were undertaken. Twenty-four OPMs were placed over the left and right sensorimotor cortex.

To generate functional images using MEG data, accurate knowledge of the location and orientation of the sensors relative to the brain is required. To facilitate this we adopted a three-step process (shown in Fig. 7):

(1) A volumetric X-ray computed tomography (CT) scan of the bicycle helmet was acquired. Fake plastic OPMs were placed in the slots to show the sensor locations and orientations relative to the helmet. This provided a 3D digitised representation of the array, at a resolution of 162.9 μm. From this digitisation, three reference points on the outer helmet surface were identified, and used to setup a co-ordinate system in which the locations and orientations of the sensors were set, relative to the helmet markers. Here, we have assumed that all OPMs are sensitive to the orientation perpendicular to the outer edge of the casing. However, the precise variation of the directional sensitivity from sensor to sensor is unknown and should be the topic of further work.

(2) Three IR retroreflective markers were placed on the helmet reference points. A further three were placed on a pair of glasses worn by the participant. The two marker sets were treated as separate rigid bodies, and their translation and rotation monitored throughout the experiment using the OptiTrack camera. This allowed knowledge of the location of the helmet relative to the glasses, and further allowed us to test whether the helmet moved, relative to the head, during the scan. In all cases, the sensors stayed within a ±5 mm distance relative to the head.

(3) Finally, the participant's head shape (scalp and face) was digitised using a 3D digitiser (Polhemus, Vermont, USA), in a co-ordinate system defined relative to the markers on the glasses. This allowed accurate knowledge of the location of the glasses relative to the head.

Combining data from the CT scan (sensor locations relative to helmet markers), the OptiTrack camera (helmet markers relative to the glasses) and the digitisation (glasses relative to the head) facilitated a complete coregistration of the location/orientation of the OPMs relative to the head. Note that mounting the fiducial markers on the head using a pair of glasses was a simple solution, which worked well. However, a potential limitation is that the glasses may move during acquisition, or the children might move them. Crucially the only requirement is three retroreflective markers at fiducial locations on the head. It is equally possible that these markers could be formed from three reflective stickers placed on the participant's face.

In order to overlay functional (MEG derived) images onto anatomical images of brain-structure, we used MRI scans. For adult participants, the anatomical image was acquired in each participant using a 3T MRI scanner running an MPRAGE[33] sequence, at an isotropic spatial resolution of 1 mm. For the children, we adopted a pseudo-MRI approach[34] whereby an existing MRI from a database was used instead of the participant's own data. Images were selected from a database containing MRIs of children in the same age group based on a best matching head shape (found using an iterative closest point algorithm).

**Experimental method: data processing**. Data were inspected visually to check for and remove excessive interference. Data from two of the OPMs for the 2-year-old participant were removed. These sensors had less opaque packaging and the OptiTrack camera's infrared LEDs produced interference on the sensor's outputs. These two sensors were replaced in the later experiments.

At the channel level we used a pseudo-gradiometer approach in which signals from neighbouring sensors were subtracted. This procedure helped to reduce residual interference. For source localisation we used a beamformer approach[35] (Fig. 7d). An estimate of the electrical source strength, $\hat{Q}_\theta(t)$, made at time $t$ and a predetermined location/orientation in the brain, $\theta$, is given by the weighted sum of sensor measurements such that

$$\hat{Q}_\theta(t) = \mathbf{W}_\theta^{\mathrm{T}} \mathbf{m}(t) \qquad (2)$$

where $\mathbf{m}(t)$ is a vector of (gradiometric) magnetic field measurements made across all sensors at time $t$, and $\mathbf{W}_\theta$ is a vector of weighting parameters tuned to $\theta$. The weighting parameters were determined using a linearly constrained minimum variance approach; mathematically

$$\min\left[\hat{Q}_\theta^2\right] \text{ subject to } \mathbf{W}_\theta^{\mathrm{T}} \mathbf{L}_\theta = 1 \qquad (3)$$

where, $\mathbf{L}_\theta$ is the forward field vector (modified to account for the gradiometer pairs used to generate $\mathbf{m}(t)$). The regularised solution to Eq. 3 is

$$\mathbf{W}_\theta^{\mathrm{T}} = [\mathbf{L}_\theta^{\mathrm{T}}\{C + \mu I\}^{-1}\mathbf{L}_\theta]^{-1}\mathbf{L}_\theta^{\mathrm{T}}\{C + \mu I\}^{-1} \qquad (4)$$

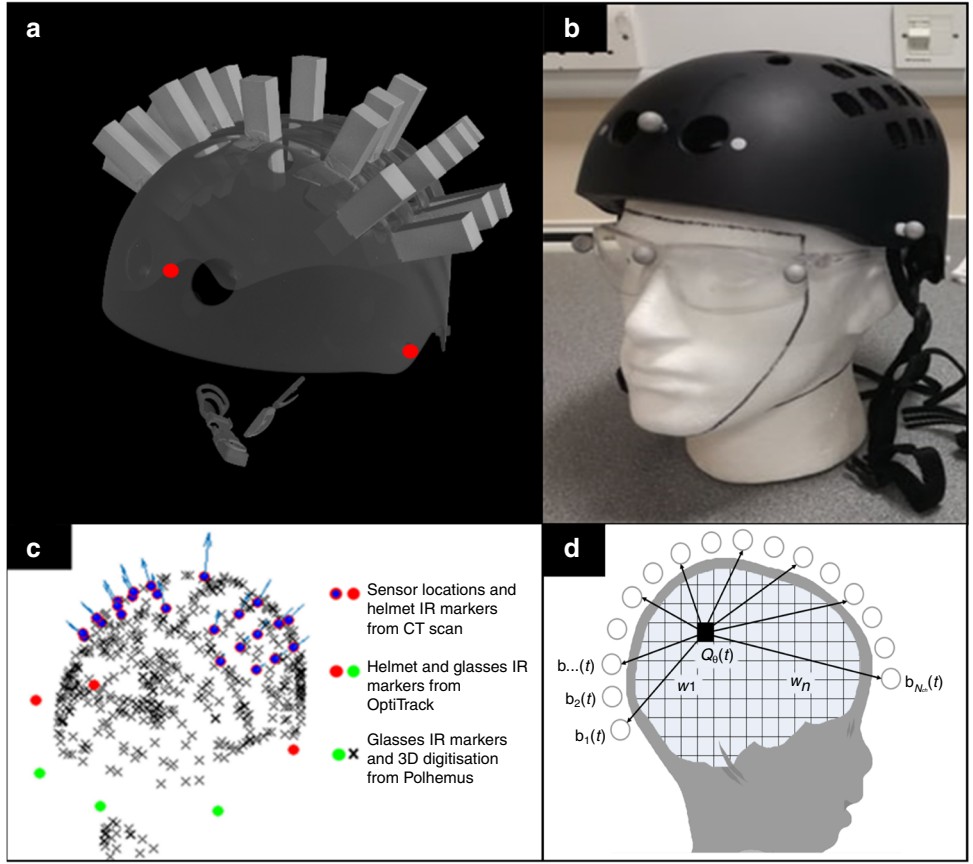

**Fig. 7** Co-registration process. **a** A volumetric X-ray CT scan of the modified bicycle helmet. Three identifiable reference points on the outer helmet surface were chosen (red circles), one on the forehead, and the others above the left and right ears. These were later used to attach reflective markers visible to an IR tracking camera. **b** A photograph of the modified bike helmet. Note the three IR reflective markers attached to the surface. The participant also wears a pair of glasses, attached to which are a further three IR reflective markers, enabling coregistration of the helmet to the head **c** 3D digitisation of the participant's head shape, defined relative to the markers on the glasses (green circles). This digitisation has been aligned to the markers on the helmet (red circles) and OPM sensors (purple markers) to provide complete coregistration of the system to brain anatomy. **d** Schematic of the beamformer technique used to localise the source of beta modulation

where $I$ is the identity matrix, $C$ represents the channel-level data covariance matrix, and $\mu$ is a regularisation parameter. Weights were determined using data covariance measured in beta band filtered (13–30 Hz) data, and using a time window capturing the entire experiment. The regularisation parameter was set to 0.05 times the maximum eigenvalue of the un-regularised matrix. The forward field was computed using a simple spherical head model and the dipole approximation first described by Sarvas[6]. In order to generate images of the spatial signature of beta modulation, two further covariance matrices were defined: $C_a$ represents data covariance in the active window ($0\,s < t < 2\,s$ relative to trial onset) and $C_c$ represents data covariance in the control window ($3\,s < t < 5\,s$). A pseudo-T-statistical contrast, given by

$$\mathrm{T}_\theta = \frac{\mathbf{W}_\theta^{\mathrm{T}} C_a \mathbf{W}_\theta - \mathbf{W}_\theta^{\mathrm{T}} C_c \mathbf{W}_\theta}{2\mathbf{W}_\theta^{\mathrm{T}} C_c \mathbf{W}_\theta} \qquad (5)$$

was computed at the vertices of a regular 4 mm grid spanning the whole brain to generate the functional images. Note that the orientation of the source in each voxel was determined as that which gave the maximum signal-to-noise ratio.

We used both sensor space (gradiometer) data and beamformed data to generate time-frequency spectrograms (TFSs). To this end, either the raw neuromagnetic signal or a source signal estimate ($\hat{Q}_\theta(t)$) was frequency filtered into overlapping frequency bands. A Hilbert transform generated the amplitude envelope of oscillations within each band. The envelope time courses were then averaged across trials, and concatenated in frequency to form a TFS showing change in oscillatory amplitude over time. For Figs. 1 and 2 the TFS shows absolute change in oscillatory amplitude from baseline, in units of fT. i.e. a baseline oscillatory amplitude was defined, individually for every frequency band, in the 3.5–4.5 s window (maternal touch task) or the 1–3 s window (Robin Hood task). These values were then subtracted from the TFS leave a measure of absolute oscillatory amplitude change relative to baseline. For Fig. 3, in order to highlight changes in gamma activity, we present relative change in oscillatory power. (Here, the baseline amplitude was defined for all bands in the 9–14 s window; this was then subtracted from the TFS, and the result divided by the same baseline, yielding a fractional measurement). For Fig. 5, we again present absolute change (baseline

taken in the 4–5 s window) but since these are source localised data, the units are nAm.

**Reporting summary**. Further information on research design is available in the Nature Research Reporting Summary linked to this article.

## Data availability
The datasets generated during and/or analysed during the current study are available from the corresponding author on reasonable request.

## Code availability
The MATLAB code used to analyse the current study is available from the corresponding author on reasonable request.

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

## Acknowledgements

This study was funded by a Wellcome Collaborative Award in Science (203257/Z/16/Z and 203257/B/16/Z) awarded to G.R.B., R.B. and M.J.B. We also acknowledge the UK Quantum Technology Hub for Sensors and Metrology, funded by the Engineering and Physical Sciences Research Council (EP/M013294/1). The Wellcome Centre for Human Neuroimaging is supported by core funding from Wellcome (203147/Z/16/Z).

## Author contributions

R.H.—research design, system fabrication, data collection, data analysis, data interpretation, writing paper. E.B.—system fabrication, data collection, data analysis, writing paper. N.H.—system fabrication, data collection, data analysis, writing paper. C.H.—research design, writing paper. Z.S.—research design, system fabrication, data collection, writing paper. J.L.—system fabrication. G.R.—system fabrication. V.S.—system fabrication. T.T.—system fabrication, writing paper. M.W.—research design, data interpretation, writing paper. C.S.—research design, data interpretation, writing paper. G.B.—system fabrication, writing paper. R.B.—research design, system fabrication, data interpretation, writing paper. R.S.—research design, data collection, data analysis, data interpretation, writing paper. M.B.—research design, system fabrication, data collection, data analysis, data interpretation, writing paper.

## Competing interests

V.S. is the founding director of QuSpin, the commercial entity selling OPM magnetometers. QuSpin built the sensors used here and advised on the system design and operation, but played no part in the subsequent measurements or data analysis. This work was funded by a Wellcome award which involves a collaboration agreement with QuSpin. All other authors declare no competing interests.
