## [Peer Review File · Nature Communications]

Reviewers' comments:

Reviewer #1 (Remarks to the Author):

This study describes the development of a pediatric OPM-MEG system. The primary innovation for this study is identifying a practical approach for the use of already commercially available OPM sensors for a child-friendly OPM array. Through simulations, multiple options for housing OPM sensors were explored ranging from a head cast system, to the adult-optimized SQUID based MEG sensors systems. It was determined that a bicycle helmet array option is fairly well optimized relative to the much more restrictive head cast system. Clear signal from the sensorimotor system during a tactile task was obtained in both a 2 and 5-year-old child. The manuscript provides high quality signal from two participants using a new much more practical OPM design for children. However, the manuscript as currently written is misleading regarding the expansion of the current data collection parameters to a whole-head design. Besides the comments listed below, sufficient detail is included in the manuscript to reproduce the results. Providing strong simulation results supporting the use of a much more child-friendly and lower cost option relative to the head-cast approach for OPM-based MEG is an important contribution to the field.

Major concerns:

1. The system is portrayed as an alternative to either EEG or NIRS. However, the data were collected with only 16 channels, despite the simulations using 81 OPM sensors. While this represents an important advance, the concern with adding sufficient sensors to rival current EEG sensor density has not been acknowledged or addressed. A primary concern related to the presentation of the manuscript is weight. Do the 16 channels represent an upper limit to what is thought to be feasible as a "wearable" array? How will this limitation be addressed for further advance of this system?
2. The main manuscript is misleading with regards to identifying what region demonstrated changes in signal in response to the tactile stimulation. It does not clearly state that the small number of sensors are only located over the sensorimotor cortex such that the only region that would be visible to the inverse algorithm would be sensorimotor cortex. While it is true that the response observed is consistent with a large body of literature and essentially exactly what is expected, the introduction does not acknowledge the limited view of the sensor array presented in this manuscript.
3. What is the estimated precision of the orientation of the OPM sensors with the use of dummy OPM sensors used during the CT vs. during actual data collection? This limitation of the current OPM subject-specific designs should be acknowledged in the manuscript.

Minor concerns:

1. What steps were taken to reduce vibration of the sensors? This statement was included in the methods but is not described.
2. How did the glasses fit on the children's heads (were they loose or fitted) and were these problematic for the children during data collection?
3. Variable "C" is defined before it is introduced in equation 3 (it is presented as if C is introduced in Eqn. 2).

Reviewer #2 (Remarks to the Author):

This study by Hill and colleagues introduces the use of Optically-pumped magnetometers (OPMs) in pediatric populations. OPMs are a rapidly emerging technology that may replace standard MEG systems, which require expensive and limited cryogenics. Standard (SQUID-based) MEGs are also very large making the technology immobile and not particularly applicable to very young children. Further, the spatial resolution of OPMs should be at least an order of magnitude better than standard MEG systems since the sensors are directly on the scalp (vs ~3 cm away). The current report includes the first data from children (a two year-old and a five year-old) using the system and follows up on a recent Nature paper by the same group, which first showed the use of spatially-resolved OPM data. Overall, I found the study interesting but did not understand the logic behind using the maternal touch task (why not something better controlled?), and was concerned

as to whether the novelty was sufficient to warrant publication in this journal. Basically, the analysis is extremely similar to the recent adult study in Nature, so does strapping a different type of helmet on a child warrant such a high impact second publication. I personally think that more is needed and hope the authors can expand the paper. Perhaps the authors could record movements in the five year old and reveal motor gamma activity or similar. Recording occipital data in both children and reporting the multispectral data would also be novel at this age and help advance the paper. Very little is known about occipital gamma in children under 8 years old.

A new tool for functional brain imaging with lifetime compliance: Response to reviewers

Ryan Hill, Elena Boto, Niall Holmes, Caroline Hartley, Zelekha Seedat, James Leggett, Gillian Roberts, Vishal Shah, Tim M. Tierney, Mark Woolrich, Charlotte Stagg, Gareth Barnes, Richard Bowtell, Rebecca Slater, and Matthew Brookes

We thank the editor and referees for their time and thoughtful comments on our paper. We have addressed these comments, point by point, with our responses in bold typeface below. Additions to our manuscript are presented here in bold italic typeface.

Reviewer #1 (Remarks to the Author):

This study describes the development of a pediatric OPM-MEG system. The primary innovation for this study is identifying a practical approach for the use of already commercially available OPM sensors for a child-friendly OPM array. Through simulations, multiple options for housing OPM sensors were explored ranging from a head cast system, to the adult-optimized SQUID based MEG sensors systems. It was determined that a bicycle helmet array option is fairly well optimized relative to the much more restrictive head cast system. Clear signal from the sensorimotor system during a tactile task was obtained in both a 2 and 5-year-old child. The manuscript provides high quality signal from two participants using a new much more practical OPM design for children. However, the manuscript as currently written is misleading regarding the expansion of the current data collection parameters to a whole-head design. Besides the comments listed below, sufficient detail is included in the manuscript to reproduce the results. Providing strong simulation results supporting the use of a much more child-friendly and lower cost option relative to the head-cast approach for OPM-based MEG is an important contribution to the field.

We thank the referee for their very positive comments. Regarding the expansion to whole head, this is addressed below.

Major concerns:

1. The system is portrayed as an alternative to either EEG or NIRS. However, the data were collected with only 16 channels, despite the simulations using 81 OPM sensors. While this represents an important advance, the concern with adding sufficient sensors to rival current EEG sensor density has not been acknowledged or addressed. A primary concern related to the presentation of the manuscript is weight. Do the 16 channels represent an upper limit to what is thought to be feasible as a “wearable” array? How will this limitation be addressed for further advance of this system?

We understand and completely agree with the referee that this important point was overlooked in our original version. We have now added the following to our discussion:

“...Most importantly, we used a limited number of OPMs which meant only a fraction of the head was covered. Clearly a complete system would require whole head coverage (similar to that simulated in Figure 4 with e.g. 81 sensors). There is no fundamental reason why the current system cannot be expanded by adding more OPMs. The only theoretical concern is cross-talk between sensors, however this is a minor consideration since at the current sensor spacing cross-talk generates only ~3% error in sensor gain¹, which can be easily accounted for in source modelling. A more practical issue is system weight. Each OPM weighs 20 g and so an 81 channel system, including the helmet, would weigh ~1800 g. Further, the cables to each OPM currently weigh 33 g/m, and cables hanging from the head cause torque which could become uncomfortable

for long experimental durations, particularly for younger participants. However, these problems are solved by the availability of a new generation of OPMs which are smaller (1.24 x 1.66 x 2.44 cm³), lighter (4 g), and use lighter cabling (3.3 g/m). This means that the weight of an 81 channel system could be as low as 500 g, with cable weight being negligible ... “

As an aside, we are now developing a 50-channel OPM-MEG system, which will hopefully have whole brain coverage, based on these next generation sensors. The principle is identical (with the sensors being held in a modified bike helmet). However, this system is not yet in operation and cannot be shown here.

2. The main manuscript is misleading with regards to identifying what region demonstrated changes in signal in response to the tactile stimulation. It does not clearly state that the small number of sensors are only located over the sensorimotor cortex such that the only region that would be visible to the inverse algorithm would be sensorimotor cortex. While it is true that the response observed is consistent with a large body of literature and essentially exactly what is expected, the introduction does not acknowledge the limited view of the sensor array presented in this manuscript.

We agree that again this point was overlooked. However, we do point out that in our validation of the source localisation (in adults), we used 24 sensors which bilaterally covered the entire sensory and motor strips, and the responses were localised to the hand area of the contralateral motor cortex. However, evidence for this was relatively limited in our original submission. To further demonstrate this, we have undertaken additional analyses. Our results section now reads:

“...To further validate this coregistration procedure, three adults underwent a finger movement paradigm¹. A single trial comprised 2 s of right index finger abduction followed by 3 s rest. The experiment consisted of 50 trials. A 24 OPM array which covered the left and right primary motor cortices, was used. Results are shown in Figure 5B. Note that, despite relatively sparse sensor coverage, beta modulation was well localised to contralateral motor cortex. Quantitatively, the peaks in the beta response were found at MNI coordinates (-44 -16 56) mm, (-36 -22 48) mm and, (-42 -6 52) mm for the three subjects. According to the Oxford-Harvard probabilistic atlas, all three coordinates fall in pre-central-sulcus, as would be expected for a motor response. In addition, all three subjects reconstructed data show the expected beta-band modulation with a clear movement related beta decrease during movement and post movement rebound on movement cessation...”

New text has been added to the Discussion, as follows:

“...Finally limited scalp coverage and low sensor count means that source localisation is intrinsically constrained to specific brain areas (i.e. we will only localise sources under the sensors). Nevertheless we have shown the expected left hemisphere lateralisation of sensory and motor responses, and in our validation (Figure 5) we see precise localisation to the hand area, in pre-central sulcus; this was with coverage using 24 sensors across the entire motor strip...”

3. What is the estimated precision of the orientation of the OPM sensors with the use of dummy OPM sensors used during the CT vs. during actual data collection? This limitation of the current OPM subject-specific designs should be acknowledged in the manuscript.

This is a very good question. The orientation of the sensor casing is obtained from the X-ray CT of the helmet. The sensors are held rigidly inside the helmet and the X-ray CT is high quality with

negligible distortions; for this reason we expect that the orientation of the sensors can be determined relative to the helmet with sub-degree accuracy.

The more challenging concern is to measure the accuracy of the orientation sensitivity of the OPM-itself. In all of our current work, we have assumed that all OPMs are sensitive to the orientation perpendicular to the outer edge of the casing. We have some anecdotal data suggesting that this is an accurate assumption to within a few degrees. However, the precise variation of the directional sensitivity, from OPM to OPM is unknown, and should be the topic of further work. We have therefore added a note to our Methods section

“...the locations and orientations of the sensors were set, relative to the helmet markers. Here, we have assumed that all OPMs are sensitive to the orientation perpendicular to the outer edge of the casing. However, the precise variation of the directional sensitivity from sensor to sensor is unknown and should be the topic of further work...”

Minor concerns:

1. What steps were taken to reduce vibration of the sensors? This statement was included in the methods but is not described.

We have now added to our methods.

“...In order to stabilise the sensors, small supporting blocks were screwed to the surface of the helmet, and the OPMs taped to these blocks. This ensured that the sensors could not move relative to the helmet during data acquisition...”

2. How did the glasses fit on the children’s heads (were they loose or fitted) and were these problematic for the children during data collection?

We agree that this is an important point. We have added to our methods:

“...mounting the fiducial markers on the head using a pair of glasses was a simple solution which worked well. However, a potential limitation is that the glasses may move during acquisition, or the children might move them. Crucially the only requirement is three retro-reflective markers at fiducial locations on the head. It is equally possible that these markers could be formed from three reflective stickers placed on the participant’s face....”

3. Variable “C” is defined before it is introduced in equation 3 (it is presented as if C is introduced in Eqn. 2).

Thank you! This has now been corrected

Reviewer #2 (Remarks to the Author):

This study by Hill and colleagues introduces the use of Optically-pumped magnetometers (OPMs) in pediatric populations. OPMs are a rapidly emerging technology that may replace standard MEG systems, which require expensive and limited cryogenics. Standard (SQUID-based) MEGs are also very large making the technology immobile and not particularly applicable to very young children. Further, the spatial resolution of OPMs should be at least an order of magnitude better than standard MEG systems since the sensors are directly on the scalp (vs ~3 cm away). The current report includes the first data from children (a two year-old and a five year-old) using the system and follows up on a recent Nature paper by the same group, which first showed the use of spatially-resolved OPM data. Overall,

I found the study interesting but did not understand the logic behind using the maternal touch task (why not something better controlled?), and was concerned as to whether the novelty was sufficient to warrant publication in this journal. Basically, the analysis is extremely similar to the recent adult study in Nature, so does strapping a different type of helmet on a child warrant such a high impact second publication. I personally think that more is needed and hope the authors can expand the paper. Perhaps the authors could record movements in the five year old and reveal motor gamma activity or similar. Recording occipital data in both children and reporting the multispectral data would also be novel at this age and help advance the paper. Very little is known about occipital gamma in children under 8 years old.

We thank the referee for their comments.

Firstly relating to novelty and impact, we feel strongly that the present paper goes significantly beyond “strapping a different helmet on a child”. Rather, the paper takes a very promising but impractical prototype (reported in our previous paper) and turns it into a practical system with lifetime compliance. This was made possible via extensive simulations and the introduction of novel techniques for coregistration. We have attempted to make these significant steps clear in our revised manuscript:

“...This communication has described the practical implementation of a generic, lifetime-compliant, OPM-MEG system. Using simulations we demonstrated that a simple, modified bicycle helmet offered a good compromise between high signal strength (getting sensors as close as possible to the head), and a practical and ergonomic design that could be deployed across all ages, and would be well tolerated by children. By introduction of a novel technique for coregistration between sensor locations and brain anatomy, we were able to generate 3-dimensional images of the spatial signature of changes in neural oscillations. We have used this system to provide a snapshot of how a single OPM-MEG device (with differently sized helmets) could be deployed to better understand neurodevelopment, in young children (aged 2 and 5), during adolescence (a 14-year-old) and in adults. Importantly, these are the first data to be collected using OPM-MEG in children and they show the feasibility of the technique to generate high quality data, even in a two year-old child (typically the hardest age to scan without sedation)...”

This said, we do agree with the referee that our demonstration of this system could have been taken much further to show some of the myriad possibilities for new types of neurodevelopmental study that could be made possible. And also to better demonstrate lifetime compliance. As suggested by the reviewer, we have therefore substantially expanded the paper to include (i) a computer game that is controlled by movement, and (ii) a natural motor learning task.

We appreciate the suggestion by the referee to measure occipital gamma activity. However, whilst this is something we intend to do in the near future, we feel it requires a proper cross-sectional study design, scanning a minimum of 15-20 subjects across a number of age brackets in order to show the evolution of the gamma band signal. Additionally, we feel we need a larger number of OPMs with better coverage (we are now awaiting the fabrication of this larger array). For this reason the gamma study suggested by the referee was deemed beyond the scope of the present study. Instead, we have chosen to stick with the sensorimotor system, and provide two further demonstrations of naturalistic stimulation, one in a child, and the other in a young adult – both relating to brain development.

In the first demonstration, we measured brain activity in a 14-year-old participant who played a simple computer game controlled by movement (e.g. analogous to Nintendo Wii). The following text has been added to our paper to describe this work:

“...we aimed to show that the same system could be used in older children, alongside a naturalistic (and engaging) motor paradigm. A single 14-year-old female participant took part in the experiment. Our “Robin Hood” paradigm (See Figure 2A) comprised a set of targets displayed on a screen. A cross hair smoothly moved across the screen at a set speed, and at specific times passed over a target. The subject had to move their right index finger in order to ‘shoot’ an arrow at the target. When a shot was fired the position of the arrow was recorded as a black marker on the screen, and the participant was able to score points by hitting the targets. We reasoned that this paradigm, which took the form of a motion-controlled computer game, would be engaging for teenagers. Control of the game via movement was enabled by an infra-red reflective marker placed on the right index finger. Movement was tracked by a 3D camera and fed back, in real time to the paradigm, as well as being recorded by the OPM-MEG system. The game slowly got harder during the experiment with the cross hairs moving more quickly.

The recorded finger movement is shown in Figure 2B; note that as the task progresses the movement required becomes faster. The finger movement can be characterised accurately, and used to trigger data analysis. In this case we synchronised data analysis (time $t = 0$) to the measured offset of motion. The sensor-level MEG results are shown in Figure 2C. In the upper plot, the line thickness indicates which gradiometer pairings showed the largest beta band response. The beta response itself is shown below in the time-frequency spectrogram, with the expected desynchronization during movement shown clearly²³...”

[redacted]

Figure 2: An interactive motor paradigm: A) The paradigm: a cross hair moves across the screen passing targets. The participant moved their finger in order to ‘shoot’ an arrow at the target. B) Measured finger movements for three different speeds of cross hair movement. The shaded area shows the average and standard deviation across trials whilst the solid line shows a single example trial. C) Beta responses in a single 14 year old participant. The upper plot shows a schematic diagram of the positions of the OPMs; the thickness of the line denotes the strength of the change in beta amplitude for all gradiometers. The lower plot shows a time frequency spectrogram from the most anterior gradiometer pair, with time zero representing the offset of movement, measured from the recorded finger movement; all trials have been averaged.

In the second demonstration, we aimed to exploit the ability to move, to demonstrate that a natural motor learning task (learning to play a musical instrument) could be undertaken in our system, whilst MEG data were recorded. The following text has been added to describe this demonstration:

“...As a final demonstration of the versatility of our system we aimed to scan an adult executing a naturalistic task. A single 24-year-old female took part in an experiment where she was asked to play a musical instrument. Specifically, they were asked to learn to play a sequence of five chords on a ukulele. The chords were visually projected onto a screen, and she was given 5 s to complete the sequence. This was repeated 40 times. The experimental set-up is shown in Figure 3A with the inset image showing the visual presentation. Figure 3B shows measured head and hand movement, as well as the recorded sound made by the instrument. A single example trial is shown (in which the subject only managed to play three of the five chords). In this naturalistic motor learning task, the participant had to make natural head movements, as they looked first at the chord pattern on the screen, and then at the frets on the ukulele to form the chord. These head movements were measured to be 4.73 ± 1.21 cm (translation) and $28.1^\circ \pm 6.71^\circ$ (rotation) (average \pm standard deviation in the dominant direction across trials). Despite the substantial head movement that was required to complete the task, clear electrophysiological responses are observed in the time frequency spectrum shown in Figure 3C, including both the expected beta²³ and gamma²⁵ band responses...”

Figure 3: Naturalistic motor learning: A) Experimental set up with a participant playing a musical instrument. Inset image shows what the participant saw on the screen. B) Measured head and right hand movement showing the timing of when the ukulele was strummed. The lower trace shows the sound generated. C) Schematic diagram of sensor placement; the line thickness denotes the strength of the effect in the gamma band (left) and the beta band (right). Time frequency spectrum shows the neural oscillatory response from the gradiometer pair indicated by the dotted red line.

We believe that the inclusion of these additional demonstrations not only demonstrates lifetime compliance, but also shows how a wearable system can be deployed in a number of different experimental paradigms to measure aspects of brain development.

REVIEWERS' COMMENTS:

Reviewer #1 (Remarks to the Author):

This is a resubmission of a manuscript describing a helmet-based system to allow for MEG data collection with an OPM sensor array across the lifespan. This new optimization is important because the authors have demonstrated that signal can be optimized across different head sizes with the use of an inexpensive bicycle helmet as a holder for the OPM sensors. They have further demonstrated the utility of the system for use with naturalistic paradigms in which the head moves during data collection. The revision has addressed my prior concerns. Only minor revisions are noted at this time.

Methods: Ukelele description: "replaces with plastic" should be "replaced with plastic"

Methods: Finger abduction description: "followed by a 3 s of baseline period" baseline is usually before the active component, to be consistent with the other descriptions this should be described as "followed by 3 s of rest"

Fig. M2B, description is awkward "... markers placed on"

I would recommend "lifespan compliance" rather than "lifetime compliance," which has connotations of device lifetime rather than human lifetime.

Results, 1st sentence: age 2 and 5 "years"

Results 1st paragraph: vapour is not singular, remove "a"

Results: It is unclear to me why Fig. 5 and the accompanying text is placed at the end of the results. Since there are other source analysis results shown earlier, it seems that this validation of the registration procedure should be presented before any of the pediatric results.

Reviewer #2 (Remarks to the Author):

The authors have responded very well to my previous concerns, especially in regard to the added value of this paper over the existing Nature paper. They have added two more experiments demonstrating the unique opportunities offered by the helmet design, including an adolescent playing a Nintendo-like game and an adult playing a ukulele. In both experiments, motor-related beta responses can be clearly seen in the spectrograms, supporting the feasibility of such experiments and the veracity of the measurements. I was also pleased to see the added detail in the coregistration process, which is critical for precise source reconstruction, and the added detail and modeling to show how different helmet designs impacted sensitivity to neural responses across the brain. Given the added experimental data, modeling, and methodological points, the added value of the paper is now very high and feel it's a major contribution to the literature and rapid growth of OPMs. That said, before publication I would like to see the authors address these minor points:

1. The authors employed an artificial gradiometerization approach, whereby they appear to have subtracted the signal of neighboring OPMs prior to plotting the spectrograms and other results. Can the authors comment in the paper on how they determined the proper polarity of the resulting signals? It would seem this would be arbitrary and based on the direction of the subtraction. Also, since the two OPMs in the subtraction are a considerable distance apart, it would seem that they may often have opposite polarity due to being on different sides of an active dipolar source. How was this handled?

2. Related to the above, can the authors comment on whether the subtracted signals were used in the source reconstruction? It would seem doing such would create major problems, but perhaps I am missing something.

3. For each experiment, the authors need to clarify the baseline and how the shown spectrogram data was adjusted based on this. For example, in Figure 3 it is a % scale and guessing it was normalized against the baseline mean per frequency bin. For the others, it is expressed as output of a Hilbert so could be simple subtraction of the baseline mean or another method. Please clarify.

4. Figure 3C: In the spectrogram, is the color scale bar's units correct? The change for beta was 0.1% to 0.4%? That seems tiny. In SQUID based systems, the change is usually 10-40% so wondering if the decimal is off.

5. Figure 4: To generate such a sensitivity plot, one would need to choose a physiological amplitude per unit space. In digging through the methods, it appears the authors used 1 nAm per location (~27,000 point dipole grid) to generate these maps. That's probably fine, but these are critical details and would add them to the results section, ideally in the text and caption. Also, please label the color scale bar in this figure so that the units are clear, and label the y-axis on the signal ratio plot in the bottom right.

6. Figure 5 caption states that the peak was located in premotor cortex in all participants. This is a bit odd (would expect precentral gyrus based on many past studies) and from the images it appears not to be in premotor cortex for at least two of the three. Please check that this is correct.

7. Since the audience at Nature Comm is broad and most will be unfamiliar with the MEG oscillatory motor literature, the citations on this could be significantly improved. All of the experiments here involve the motor system and rely on the beta and motor gamma responses, which have been widely characterized. Surprisingly, the authors only cite the 1999 review paper by Pfurtscheller and a recent gamma paper by Gaetz. The 1999 review was a landmark paper, but this review is almost entirely EEG based and the way beta oscillations are understood now is a major departure from that presented by Pfurtscheller. Our basic understanding of what the responses mean and the parameters that affect them have changed drastically over the past 20 years, and this should be reflected here. Would recommend the authors include additional recent citations from some of the leaders in MEG studies of motor-related oscillations (e.g., Heinrichs-Graham, Cheyne, and potentially other papers from Gaetz).

Tony W Wilson, PhD

A new tool for functional brain imaging with lifespan compliance: Response to reviewers

Ryan Hill, Elena Boto, Niall Holmes, Caroline Hartley, Zelekha Seedat, James Leggett, Gillian Roberts, Vishal Shah, Tim M. Tierney, Mark Woolrich, Charlotte Stagg, Gareth Barnes, Richard Bowtell, Rebecca Slater, and Matthew Brookes

We thank the editor and referees for their time and thoughtful comments on our paper and we are delighted to receive our provisional acceptance. We have addressed the referees remaining comments, point by point, below. Our responses in bold typeface below. Additions to our manuscript are presented here in bold italic typeface.

Reviewer 1:

Methods: Ukulele description: "replaces with plastic" should be "replaced with plastic"

Thank you. This typo has now been corrected.

Methods: Finger abduction description: "followed by a 3 s of baseline period" baseline is usually before the active component, to be consistent with the other descriptions this should be described as "followed by 3 s of rest"

We agree and this has been changed.

Fig. M2B, description is awkward "... markers placed on"

We agree that the caption for Figure M2 was clumsy and it has now been amended as follows:

Figure M2: Co-registration process. A) A volumetric X-ray CT scan of the modified bicycle helmet. Three identifiable reference points on the outer helmet surface were chosen (red circles), one on the forehead, and the others above the left and right ears. These were later used to attach reflective markers visible to an IR tracking camera. B) A photograph of the modified bike helmet. Note the three IR reflective markers attached to the surface. The participant also wears a pair of glasses, attached to which are a further three IR reflective markers, enabling coregistration of the helmet to the head C) 3D digitisation of the participant's head shape, defined relative to the markers on the glasses (green circles). This digitisation has been aligned to the markers on the helmet (red circles) and OPM sensors (purple markers) to provide complete coregistration of the system to brain anatomy. D) Schematic of the beamformer technique used to localise the source of beta modulation.

I would recommend "lifespan compliance" rather than "lifetime compliance," which has connotations of device lifetime rather than human lifetime.

We agree and this has been changed

Results, 1st sentence: age 2 and 5 "years"

This has been changed

Results 1st paragraph: vapour is not singular, remove "a"

Thank you. This has been changed

Results: It is unclear to me why Fig. 5 and the accompanying text is placed at the end of the results. Since there are other source analysis results shown earlier, it seems that this validation of the registration procedure should be presented before any of the pediatric results.

We understand the comment and to an extent agree. However, we wrote the paper so as to emphasise the main results first, and the technical innovations second. We did it this way in the hope that we would maximise both impact and interest for the broad readership of Nature Communications. However, this said, we are more than happy to switch the order of the paper as the referee suggests, if the editor thinks that this would be more appropriate. We leave it to the discretion of the editor.

Reviewer 2:

1. The authors employed an artificial gradiometerization approach, whereby they appear to have subtracted the signal of neighboring OPMs prior to plotting the spectrograms and other results. Can the authors comment in the paper on how they determined the proper polarity of the resulting signals? It would seem this would be arbitrary and based on the direction of the subtraction. Also, since the two OPMs in the subtraction are a considerable distance apart, it would seem that they may often have opposite polarity due to being on different sides of an active dipolar source. How was this handled?

The gradiometer approach that we used is a classical planar gradiometer approach used in the design of a number of cryogenic MEG systems. As the referee points out, the signals are likely to have opposite polarity; indeed this is the point since the neuromagnetic signal is amplified at the expense of spatial resolution. The signal polarity is then simply accounted for in the lead field matrix used in the beamformer formulation. The referee is correct that the output polarity of the beamformer is arbitrary, but that is the case for all beamformer outputs (whether based on gradiometers or magnetometers). Of course, the actual results presented are not raw signals, but rather are oscillatory envelopes derived from the Hilbert Transform – there is therefore no sign ambiguities in the final analyses.

2. Related to the above, can the authors comment on whether the subtracted signals were used in the source reconstruction? It would seem doing such would create major problems, but perhaps I am missing something.

As noted above, the gradiometer signals were used in the analysis, and accounted for in the lead field matrix (similar to how beamforming is used on a conventional (Elekta) system). We have now made this clear in the methods.

"...An estimate of the electrical source strength, $Q_{\theta}(t)$, made at time t and a predetermined location/orientation in the brain, θ , is given by the weighted sum of sensor measurements such that

$$\hat{Q}_\theta(t) = W_\theta^T m(t) \quad [2]$$

where $m(t)$ is a vector of (gradiometric) magnetic field measurements made across all sensors at time t , and W_θ is a vector of weighting parameters tuned to θ . The weighting parameters were determined using a linearly constrained minimum variance approach; mathematically

$$\min[\hat{Q}_\theta^2] \text{ subject to } W_\theta^T L_\theta = 1 \quad [3]$$

where, L_θ is the forward field vector (modified to account for the gradiometer pairs used to generate $m(t)$)..."

3. For each experiment, the authors need to clarify the baseline and how the shown spectrogram data was adjusted based on this. For example, in Figure 3 it is a % scale and guessing it was normalized against the baseline mean per frequency bin. For the others, it is expressed as output of a Hilbert so could be simple subtraction of the baseline mean or another method. Please clarify.

We apologise for not making this clear. In Figures 1 and 2, we show the oscillatory envelopes of raw gradiometer data measured in fT. In Figure 3 we show relative change – this was in order to better delineate the gamma band response (which is otherwise swamped by the beta signal if shown as an absolute change). In Figure 5 we show source localised data (in units of nAm). This has now been made clear in the text.

"...We used both sensor space (gradiometer) data and beamformed data to generate time-frequency spectrograms (TFSs). To this end, either the raw neuromagnetic signal or a source signal estimate ($Q_\theta(t)$) was frequency filtered into overlapping frequency bands. A Hilbert transform generated the amplitude envelope of oscillations within each band. The envelope time courses were then averaged across trials, and concatenated in frequency to form a TFS showing change in oscillatory amplitude over time. For Figures 1 and 2 the TFS shows *absolute* change in oscillatory amplitude from baseline, in units of fT. I.e. a baseline oscillatory amplitude was defined, individually for every frequency band, in the 3.5 to 4.5 s window (maternal touch task) or the 1 to 3 s window (Robin Hood task). These values were then subtracted from the TFS leave a measure of absolute oscillatory amplitude change relative to baseline. For Figure 3, in order to highlight changes in gamma activity, we present *relative* change in oscillatory power. (Here, the baseline amplitude was defined for all bands in the 9 to 14 s window; this was then subtracted from the TFS, and the result divided by the same baseline, yielding a fractional measurement). For Figure 5, we again present absolute change (baseline taken in the 4 to 5 s window) but since these are source localised data, the units are nAm..."

4. Figure 3C: In the spectrogram, is the color scale bar's units correct? The change for beta was 0.1% to 0.4%? That seems tiny. In SQUID based systems, the change is usually 10-40% so wondering if the decimal is off.

We apologise for the error. It should have said fractional change. This has now been corrected

5. Figure 4: To generate such a sensitivity plot, one would need to choose a physiological amplitude per unit space. In digging through the methods, it appears the

authors used 1 nAm per location (~27,000 point dipole grid) to generate these maps. That's probably fine, but these are critical details and would add them to the results section, ideally in the text and caption. Also, please label the color scale bar in this figure so that the units are clear, and label the y-axis on the signal ratio plot in the bottom right.

We agree and these details have been added.

"...Figure 4: Simulated MEG helmet designs for a 2-year-old. Main figures show normalised representation of "measurable signal" across the brain. (Specifically a 1nAm source dipole has been simulated at all cortical locations and we measured the Frobenius norm of the resulting sensor level field pattern. These values were normalised by the average Frobenius norm when sensors are placed on the scalp surface – i.e, the values, which mostly scale between 0 and 1, represent signal quality relative to the bespoke helmet). Inset figures show the location of simulated MEG sensors. A) Bespoke helmet. B) Helmet to fit 95% of 2-year-olds. C) Modified bike helmet. D) Cryogenic MEG system. E) Signal ratio distribution for each dipole location with respect to the bespoke helmet..."

6. Figure 5 caption states that the peak was located in premotor cortex in all participants. This is a bit odd (would expect precentral gyrus based on many past studies) and from the images it appears not to be in premotor cortex for at least two of the three. Please check that this is correct.

We apologise, this was a mistake. The peak is in precentral gyrus; motor cortex. This has now been corrected.

7. Since the audience at Nature Comm is broad and most will be unfamiliar with the MEG oscillatory motor literature, the citations on this could be significantly improved. All of the experiments here involve the motor system and rely on the beta and motor gamma responses, which have been widely characterized. Surprisingly, the authors only cite the 1999 review paper by Pfurtscheller and a recent gamma paper by Gaetz. The 1999 review was a landmark paper, but this review is almost entirely EEG based and the way beta oscillations are understood now is a major departure from that presented by Pfurtscheller. Our basic understanding of what the responses mean and the parameters that affect them have changed drastically over the past 20 years, and this should be reflected here. Would recommend the authors include additional recent citations from some of the leaders in MEG studies of motor-related oscillations (e.g., Heinrichs-Graham, Cheyne, and potentially other papers from Gaetz).

We agree. The reason for the omission was that we were originally trying to reduce word count. We have now added the references mentioned.